# Grazing-Induced Changes in circRNAs, miRNAs and mRNAs Expression in Tibetan Sheep Biceps Femoris

**DOI:** 10.3390/biology14091143

**Published:** 2025-08-29

**Authors:** Xiong Ma, Shaobin Li, Zhanzhao Chen, Zhaohua He, Jianming Ren, Shiyu Tao, Lan Zhang, Pengfei Zhao

**Affiliations:** 1Faculty of Chemistry and Life Sciences, Gansu Minzu Normal University, Hezuo 747000, China; 0700013@gnun.edu.cn (X.M.); 0700085@gnun.edu.cn (J.R.); 0700100@gnun.edu.cn (S.T.); 0700089@gnun.edu.cn (L.Z.); 2Gansu Key Laboratory of Herbivorous Animal Biotechnology, Faculty of Animal Science and Technology, Gansu Agricultural University, Lanzhou 730070, China; lisb@gsau.edu.cn (S.L.); 1073324010058@st.gsau.edu.cn (Z.C.); 3College of Biology and Food, Shangqiu Normal University, Shangqiu 476000, China; hezh@st.gsau.edu.cn

**Keywords:** Tibetan sheep, grazing, biceps femoris, intramuscular fat, competing endogenous RNA

## Abstract

The transition from natural grazing to pen-feeding may potentially compromise the meat quality of Tibetan sheep, a crucial livestock species on the Tibetan Plateau. This study assesses the impact of grazing on meat quality through a comparative analysis of the biceps femoris in grazing and pen-fed sheep. The results demonstrate that grazing significantly enhances intramuscular fat content, potentially improving the meat’s tenderness and juiciness. Further analysis reveals alterations in the expression of circRNAs and miRNAs within muscle tissue, which may influence muscle fiber type and energy metabolism processes. These findings suggest that grazing could enhance meat quality by promoting a shift towards muscle fibers with increased fat storage capacity. The research provides valuable insights for augmenting the quality of Tibetan sheep meat and could contribute to the development of efficient meat quality control strategies, ultimately benefiting local communities and the livestock industry.

## 1. Introduction

The Tibetan sheep, having migrated to the Tibetan Plateau from northern China approximately 3100 years ago via the historical Tang-Bo Ancient Road, have become a permanently established breed [1]. They are classified as one of the three major primitive sheep breeds in China, alongside the Kazakh and Mongolian breeds [2,3]. Currently, the Tibetan sheep constitute the most numerous livestock species on the Tibetan Plateau and are the largest sheep breed in China by population size [2]. This prominence positions them as a vital resource for the development of the animal husbandry industry on the Tibetan Plateau.

Recent policy initiatives, including grazing prohibition and the establishment of a grazing moratorium, have led to a shift from traditional grazing to pen-feeding systems in Tibetan sheep husbandry. The assessment of meat quality encompasses various parameters, including edible attributes, nutritional content, processing characteristics, and hygiene standards. The edible quality is particularly focused on intramuscular fat (IMF) content, meat color, water-holding capacity, shear force, and cooking loss in animal skeletal muscles [4]. Skeletal muscle, which serves as the primary edible portion of livestock, generally constitutes 40–50% of their total body weight [5,6]. Studies have demonstrated that reduced physical activity associated with pen-feeding can negatively impact the quality of skeletal muscle in livestock and poultry, affecting attributes such as meat color, water-holding capacity, and cooking loss [7,8].

Skeletal muscle is composed of muscle fibers, which are distinct in their structure, function, and metabolic properties [9]. In ruminant species, these fibers are classified into three categories based on myosin ATPase activity and their metabolic profiles: Type I (slow oxidative), Type IIa (fast oxidative), and Type IIx (fast oxidative-glycolytic) [10,11]. Type I fibers are characterized by a high capacity for oxidative metabolism and reliance on aerobic processes, conferring resistance to fatigue. Type IIa fibers possess a dual metabolic capacity, enabling a balance between endurance and power generation. In contrast, Type IIx fibers primarily utilize glycolytic metabolism, allowing for rapid energy production but with reduced fatigue resistance [12]. Studies reveal a positive correlation between the proportion of Type I muscle fibers and IMF content [13], with increased IMF enhancing meat’s flavor, juiciness, and tenderness [14]. Research also indicates that variations in muscle fiber composition influence meat tenderness, with a higher proportion of Type I fibers generally associated with improved tenderness [12]. Conversely, a higher proportion of Type II muscle fibers can result in meat that is dry and tough [15].

The proportion of skeletal muscle fibers is a pivotal factor in determining meat quality, with physical activity being essential in modulating fiber type transformation [7]. Nevertheless, the effects of pen-feeding on the meat quality of Tibetan sheep, as compared to grazing, are not yet fully understood. Therefore, this study involved sampling the biceps femoris from Tibetan sheep reared under two distinct feeding regimes to evaluate meat quality, with a specific focus on IMF. Transcriptomic sequencing was employed to analyze the underlying mechanisms of meat quality alterations. By integrating phenotype and transcriptome data, the study aimed to elucidate the impact of grazing on meat quality, thereby providing foundational knowledge for the development of effective meat quality regulation strategies.

## 2. Materials and Methods

### 2.1. Experimental Animals and Sample Collection

In this study, we divide the neutered three-year-old Tibetan ram sheep from Xiahe County, Gannan Tibetan Autonomous Prefecture, Gansu Province, China, into two distinct groups: a grazing group (group G) and a control group (group C). The group G of Tibetan sheep traveled 20 km between their pen and the pasture daily, maintaining this exercise pattern for 6 months. In contrast, the group C of Tibetan sheep remained in a confined pen, with limited space for movement. To ensure the accuracy of the experimental results, the dietary structure of both groups of Tibetan sheep was kept consistent. That is, the food for group C was sourced from the forage of the same pasture, and they had free access to it. The only difference was that the group G engaged in regular long-distance round-trip activities, while the group C remained in a confined state. Each group consists of 5 biological replicates. These sheep were humanely euthanized using traditional Tibetan practices. Immediately afterwards, the biceps femoris is dissected and collected from the left side of each carcass, and rapidly stored in liquid nitrogen for subsequent IMF content analysis and transcriptome sequencing.

### 2.2. IMF Content Measurements

Using the Soxhlet extraction technique with petroleum ether as the solvent, the IMF content in the biceps femoris of 10 Tibetan sheep from groups G and C was quantitatively determined. The sample preparation steps were as follows: Firstly, the muscle samples preserved in liquid nitrogen were freeze-dried in a freeze dryer (Labconco, Kansas, KS, USA) for 48 h until completely dehydrated. Subsequently, a tissue grinder (Bertin Technologies, Montigny-le-Bretonneux, France) was used to grind the freeze-dried samples into a uniform fine powder. Approximately 5 g of muscle powder sample was precisely weighed, wrapped in filter paper, and placed in the Soxhlet extractor (Thermo Fisher Scientific, Waltham, MA, USA). The extraction process lasted for 8 h to ensure complete extraction of fat. The final measurement results were expressed as a percentage of the weight of wet muscle tissue, with each sample repeated three times [16]. SPSS 25.0 (SPSS, Chicago, IL, USA) was used for *t*-test analysis of IMF content, and a *p*-value < 0.05 indicated a significant difference.

### 2.3. RNA Extraction, Library Construction and Sequencing

Total RNA was extracted from the biceps femoris tissues of 5 Tibetan sheep from each of groups G and C and purified using TRIzol kit (Invitrogen, Carlsbad, CA, USA), with the protocol meticulously adhering to the manufacturer’s guidelines. A 1% agarose gel electrophoresis was employed to detect total RNA contamination or degradation. The assessment of total RNA purity and concentration was conducted using the Nanodrop 2000 (Thermo Fisher Scientific, Waltham, MA, USA) and the Qubit^®^ 2.0 fluorometer (Life Technologies, Carlsbad, CA, USA), respectively. The RNA integrity number (RIN) was determined via the Agilent 2100 Bioanalyzer (Agilent Technologies, Palo Alto, CA, USA), serving as a critical metric for RNA quality. For the subsequent experimental procedures, only RNA samples exhibiting high quality, with concentrations exceeding 80 ng/µL, a purity range of 1.80 to 2.10, and a RIN value greater than 7 were deemed suitable for use. The rRNA was efficiently removed from RNA samples utilizing the Ribo-Zero Gold rRNA Removal Kit (Illumina, San Diego, CA, USA). The remaining RNA was used to synthesize complementary DNA (cDNA) libraries by the NEBNext Ultra RNA Library Prep Kit (New England Biolabs, Ipswich, MA, USA). Subsequently, the constructed cDNA libraries were then subjected to paired-end sequencing by Gene Denovo Biotechnology Co., Ltd. (Guangzhou, China).

In addition, RNA molecules ranging from 18 to 30 nucleotides (nt) in length were concentrated using polyacrylamide gel electrophoresis, and the 3′ and 5′ adaptors were ligated independently. After which, reverse transcription and PCR amplification of small RNAs were conducted to link nucleotides to both ends of the adaptors. Next, PCR products of about 140 base pairs (bp) were recovered and purified for the construction of cDNA libraries by the NEBNext^®^ Multiplex Small RNA Library Prep Set for Illumina (New England Biolabs, Ipswich, MA, USA). The constructed libraries were evaluated for fidelity and yield utilizing the Agilent 2100 Bioanalyzer (Agilent Technologies, Palo Alto, CA, USA). Finally, small RNA sequencing was performed by Gene Denovo Biotechnology Co., Ltd. (Guangzhou, China).

### 2.4. Read Mapping and Transcript Assembly

Utilizing fastp v.0.23.2 (HaploX Biotechnology, Shenzhen, China), we successfully procured high-quality clean reads after excluding unqualified reads, namely those harboring adapters, reads where more than 50% of the bases have a quality score Q ≤ 20, and sequences containing in excess of 10% unknown nucleotides [17]. Subsequently, these refined reads were aligned to the *Ovis aries* reference genome (ARS-UI_Ramb_v3.0) utilizing HISAT2 v.2.2.1 (Johns Hopkins University, Baltimore, MD, USA), with the default parameters in effect [18]. Finally, the aligned reads for each sample were assembled through the StringTie v.3.0.0 (Johns Hopkins University, Baltimore, MD, USA) [19].

### 2.5. Identification of circRNAs and miRNAs

After discarding low-quality reads, the remaining high-quality clean reads are mapped to the *Ovis aries* reference genome by HISAT2 v.2.2.1 [18]. The unmapped reads were then collected, and 20 mers from both ends were extracted and aligned to the reference genome to find unique anchor positions within splice sites. Anchor reads that aligned in the reversed orientation (head-to-tail) indicated circRNA splicing and were subjected to CIRIquant (Computational Genomics Lab, Beijing, China) to identify circRNAs [20]. A candidate circRNA was called if it was supported by at least two unique back-spliced reads in at least one sample.

Subsequently, the clean reads are aligned to the miRBase database “http://www.mirbase.org/ (accessed on 2 September 2024)” to pinpoint existing *Ovis aries* miRNAs [21]. Since some miRNA sequences of *Ovis aries* have not yet been included in the miRBase database, it is necessary to align their miRNAs with those of other species to identify known miRNAs. Finally, utilizing the miRDeep2 (Max Delbrück Center for Molecular Medicine, Berlin, Germany) software, novel miRNAs are predicted based on the distinctive hairpin structures of miRNA precursors [22].

### 2.6. Quantification and Differential Expression Analysis

To accurately assess the expression levels of these annotated circRNAs, the raw counts are normalized by calculating reads per million mapped reads. The abundance of these miRNAs is normalized using the transcripts per million, adhering to established criteria. Furthermore, raw counts were used as input for the DESeq2 v.2.0 software (European Molecular Biology Laboratory, Heidelberg, Germany), which performs its own normalization. Differentially expressed circRNAs (DEcircRNAs) and differentially expressed miRNAs (DEmiRNAs) were identified using the software’s built-in methods with the following criteria: |log2(fold change)| ≥ 1 and *p*-value < 0.05 [23].

### 2.7. Functional Annotation and Pathways Enrichment Analysis

Prediction of DEmiRNA target genes using miRanda v.3.3a “http://www.microrna.org (accessed on 2 September 2024)” and Targetscan v.7.0 “https://www.targetscan.org/vert_70/ (accessed on 3 September 2024)”. The intersection of the two software results was taken as the final outcome. The parameters of miRanda v.3.3a were as follows: the score threshold is 140, the energy threshold is −10 kcal/mol, demand strict 5′ seed pairing, the gap-open penalty is −4.0, and the gap-extend penalty is −9.0. The parameters of Targetscan v.7.0 were as follows: the 2–8 nt sequences which start from 5′ small RNA were chosen as seed sequences to predict with 3′ UTR of transcripts. Subsequently, the source genes of DEcircRNAs and the candidate target genes of DEmiRNAs were subjected to Gene Ontology (GO) term “http://www.geneontology.org/ (accessed on 3 September 2024)” annotation analysis and Kyoto Encyclopedia of Genes and Genomes (KEGG) pathway “http://www.genome.jp/kegg/ (accessed on 3 September 2024)” enrichment analysis, using the cluster-Profiler package 4.16.0 (Jinan University, Guangzhou, China) in R [24]. In this analysis, a cutoff value of 0.05 was established as the threshold for determining the significant categories [25,26].

### 2.8. Construction of the circRNA-miRNA-mRNA Network

In conjunction with our existing mRNA data, RNA pairs with Spearman rank correlation coefficients (SCCs) less than −0.5 are considered as negatively co-regulated circRNA-miRNA or miRNA-mRNA pairs. While circRNA and mRNA with Pearson correlation coefficients (PCCs) greater than 0.7 are regarded as co-regulated circRNA-mRNA pairs. When the *p*-value is less than 0.05, the circRNA and mRNA, along with their shared miRNA sponges, are considered as the final competitive endogenous RNAs (ceRNA) pairs. We considered that the significant nodes located in the regulated networks are associated with changes in meat quality.

## 3. Results

### 3.1. Differences in Meat Quality

The effect of grazing on the IMF of the biceps femoris of Tibetan sheep is shown in Figure 1. In comparison to group C, IMF was significantly higher in group G (*p* < 0.05), with an increase of 0.38%.

### 3.2. Characteristics of circRNAs

In this study, a total of 53,868 circRNAs were identified in the biceps femoris tissues of Tibetan sheep from two feeding modes. Among them, 17,916 and 17,671 circRNAs were identified in groups C and G, respectively, and 18,281 circRNAs were co-expressed in two groups (Figure 2A). Based on their source gene’s location, the circRNAs identified in the biceps femoris tissues were widely distributed in the Tibetan sheep chromosomes, with the highest proportion observed in chromosomes 1–3 (Figure 2B). In addition, circRNAs up to 1000 bp in length accounted for the majority (Figure 2C). Among the different types of circRNAs, annot-exon was the most common sequence (78.65%), followed by exon-intron (11.94%) and one-exon (3.33%) circRNAs (Figure 2D).

### 3.3. Characteristics of miRNAs

An average of 11,172,151 and 11,581,767 raw reads were obtained from the biceps femoris tissue samples of C and G groups, respectively. After low-quality and unqualified reads were removed, an average of 10,617,629 and 10,917,712 clean reads were obtained from the C and G libraries, of which 79.62% and 81.51% clean reads were mapped to the sheep reference genome, respectively.

Among these reads, most small RNAs had a length of 21–23 nt, and the most common length for small RNAs was 22 nt, accounting for 35.93% and 35.81% of the total reads from C and G groups, respectively (Figure 3A). In addition, the known and existing miRNAs constituted the main components of the small RNA reads from C and G groups, at 70.00% and 67.48%, respectively (Figure 3B). Finally, a total of 608 miRNAs were identified in the biceps femoris tissues of Tibetan sheep from two feeding modes. Among the identified miRNAs, 561 were co-expressed, whereas only 25 and 22 miRNAs were expressed in the C and G groups, respectively (Figure 3C).

### 3.4. Differential Expression of circRNA and miRNA

We compared the Tibetan sheep biceps femoris tissue samples collected from the C and G groups to evaluate the different circRNAs and miRNAs expression levels. Of the 53,868 circRNAs identified, 627 were differentially expressed, of which 298 were up-regulated and 329 were down-regulated (Figure 4A,B). Among 608 miRNAs identified in the biceps femoris tissues of Tibetan sheep, 35 miRNAs showed differential expression between the C and G groups, of which 16 were up-regulated and 19 were down-regulated (Figure 4C,D).

### 3.5. GO and KEGG Analyses of Source Genes of DEcircRNAs

Results of the GO term annotation analyses showed that the source genes of DEcircRNAs between C and G groups were significantly concentrated in 84 cellular components, 68 molecular functions and 313 biological process terms. These terms primarily involve intracellular part (GO:0044424), protein binding (GO:0005515), cellular component organization (GO:0016043), and organelle organization (GO:0006996). Meanwhile, some terms were closely associated with muscle development and meat quality traits, including sarcomere (GO:0030017), myofibril (GO:0030016), muscle myosin complex (GO:0005859), ATP binding (GO:0005524), protein kinase activity (GO:0004672), microtubule motor activity (GO:0003777), anatomical structure development (GO:0048856), cellular metabolic process (GO:0044237), and regulation of muscle adaptation (GO:0043502) (Figure 5A).

The KEGG pathway enrichment analyses indicated that the source genes of DEcircRNAs between C and G groups were significantly enriched in 70 pathways. These pathways were primarily involved in the MAPK signaling pathway (ko04010), Wnt signaling pathway (ko04310), HIF-1 signaling pathway (ko04066), and pentose phosphate pathway (ko00030) (Figure 5B).

### 3.6. GO and KEGG Analyses of Target Genes of DEmiRNAs

Results of the GO term annotation analyses showed that the target genes of DEmiRNAs between C and G groups were significantly concentrated in 554 cellular components, 586 molecular functions and 4061 biological process terms. Among these terms, those closely associated with muscle development and meat quality traits were identified, such as myofibril (GO:0030016), contractile fiber part (GO:0044449), striated muscle myosin thick filament (GO:0005863), protein kinase activity (GO:0004672), collagen binding (GO:0005518), ATP binding (GO:0005524), muscle tissue development (GO:0060537), muscle cell differentiation (GO:0042692), and regulation of muscle adaptation (GO:0043502) (Figure 6A).

The KEGG pathway enrichment analyses indicated that the target genes of DEmiRNAs between C and G groups were significantly enriched in 171 pathways. These pathways were primarily involved in the MAPK signaling pathway (ko04010), Notch signaling pathway (ko04330), VEGF signaling pathway (ko04370), Wnt signaling pathway (ko04310), and HIF-1 signaling pathway (ko04066) (Figure 6B).

### 3.7. Network Diagram of circRNA-miRNA-mRNA Regulatory Relationships

We predicted the target mRNA of 35 DEmiRNAs between groups C and G and found that a total of 171 DEmRNAs were targeted and constituted 412 miRNA-mRNA pairs with SCCs less than −0.5. Subsequently, the target circRNAs of the 35 DEmiRNAs were predicted and found to target a total of 469 DEcircRNAs and constitute 454 circRNA-miRNA pairs with SCCs less than −0.5. Based on this, a total of 122 ceRNA pairs with PCCs > 0.7 and *p*-value < 0.05 were filtered. These ceRNA pairs were visualized using a Sankey diagram, and the result is shown in Figure 7.

## 4. Discussion

The current research employed transcriptome sequencing to elucidate the effects of grazing on the expression of circRNAs, miRNAs, and mRNAs in the skeletal muscles of Tibetan sheep, and to examine the relationship between these alterations and meat quality. The data indicated a significant increase in the IMF content in the skeletal muscles of grazing sheep (Figure 1). Since the IMF significantly influences meat’s flavor, juiciness, and tenderness [14], therefore, grazing may improve meat quality by increasing IMF content. The research discovered a positive correlation between the proportion of Type I muscle fiber and IMF content [13], and exercise can encourage a transition in muscle fiber types from Type II to Type I [27]. Suggesting that grazing might increase IMF by facilitating a shift in muscle fiber types.

circRNAs have been identified as significant regulators of skeletal muscle quality and fiber transformation [28,29,30,31,32]. miRNA can bind to and inhibit the expression of key mRNAs associated with muscle growth and development, thereby impacting muscle fiber type and growth [33]. The findings of our transcriptome sequencing analysis indicated that grazing significantly modulates the expression of circRNA and miRNA in the skeletal muscle of Tibetan sheep (Figure 4). GO term annotation and KEGG pathway enrichment analysis revealed that the DEcircRNAs were predominantly involved in muscle fiber transformation and energy metabolism, with their source genes enriched in the MAPK, Wnt, and HIF-1 signaling pathways (Figure 5). Furthermore, the target genes of DEmiRNAs were also primarily enriched in pathways related to muscle development and quality, such as the MAPK, VEGF, Wnt, and HIF-1 signaling pathways (Figure 6). The MAPK signaling pathway can regulate the proliferation, differentiation, and apoptosis of muscle cells, thereby affecting muscle fiber type and muscle growth [34,35]. The Wnt signaling pathway can regulate the growth, metabolism, and regeneration of muscle cells, thereby affecting muscle growth and muscle quality [36,37]. The HIF-1 signaling pathway can regulate the hypoxia adaptation of muscle cells and stimulate the expression of type I myosin [38]. The VEGF signaling pathway can stimulate the proliferation and migration of vascular endothelial cells, promoting the formation of new blood vessels, ensuring O_2_ supply, and increasing the aerobic capacity of muscle fibers [39]. The findings suggest that circRNAs and miRNAs may be integral to the molecular mechanisms regulating meat quality through grazing, as mediated by the discussed pathways.

circRNA’s role as a molecular sponge enables it to competitively bind with miRNA, thereby mitigating miRNA’s inhibitory impact on target mRNA [40]. Consequently, this research delineated a circRNA-miRNA-mRNA interaction network. The data indicated that novel_circ_001331, novel_circ_012918, and novel_circ_029843 ameliorated the inhibition of COL8A1 by interacting with miR-381-y, miR-7144-x, novel-m0040-3p, and novel-m0092-5p. Similarly, novel_circ_012918 and novel_circ_029843 alleviated the inhibition of MYLK3 through binding with miR-16-z, miR-8159-x, novel-m0092-5p, novel-m0040-3p, and oar-miR-329a-3p. Additionally, novel_circ_001331, novel_circ_029843, and novel_circ_059962 lessened the inhibition of NOX4 by interacting with miR-381-y, novel-m0040-3p, and novel-m0092-5p (Figure 7). The above-mentioned novel_circ_001331, novel_circ_012918, novel_circ_029843, and novel_circ_059962 were all significantly up-regulated in group G. COL8A1 is a major component of the extracellular matrix, providing structural support for muscle fibers and maintaining tissue integrity [41]. MYLK3 can phosphorylate myosin light chains, thereby regulating the motor activity of myosin [42]. NOX4 can catalyze the production of ROS, which can activate various signaling pathways, such as MAPK, and thus affect the proliferation and differentiation of muscle fibers [43]. Moreover, ROS can also regulate calcium ion homeostasis and promote normal muscle contraction [44]. In addition, ROS are also involved in the angiogenesis process, driving angiogenesis by promoting VEGF signaling and cell migration [45], which helps supply O_2_ to muscle tissues. This indicates that the aforementioned circRNA-miRNA-mRNA interaction network plays an important role in the molecular mechanism of grazing regulating the skeletal muscle quality of Tibetan sheep.

This research has elucidated the effects of grazing on skeletal muscle quality and the associated molecular mechanisms, providing a theoretical framework for enhancing the meat quality of Tibetan sheep. Future studies should aim to delineate the specific roles of circRNA and miRNA in the grazing-mediated regulation of meat quality and assess their potential as therapeutic targets for improving Tibetan sheep meat quality.

## 5. Conclusions

The practice of grazing has been observed to significantly elevate the IMF content in the biceps femoris muscle of Tibetan sheep. The proposed mechanism for this improvement in meat quality is through the modulation of a circRNA-miRNA-mRNA regulatory network, which in turn affects muscle fiber type conversion and energy metabolism pathways. This research provides a theoretical framework for enhancing the meat quality of Tibetan sheep and offers strategic insights for the development of robust meat quality control technologies. For instance, the study suggests that moderate exercise regimes could be implemented to facilitate IMF accumulation, and the RNAs associated with meat quality identified in this research could be employed as molecular markers for the selective breeding of superior livestock.

## Figures and Tables

**Figure 1 biology-14-01143-f001:**
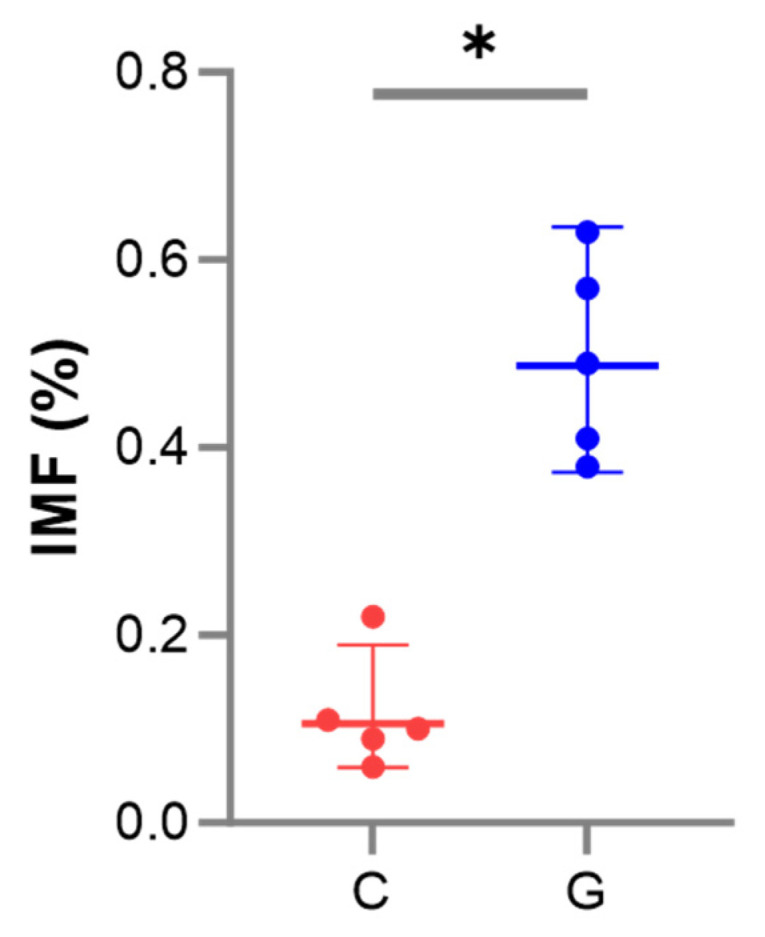
Differences in intramuscular fat (IMF) of the biceps femoris of Tibetan sheep in groups C (control) and G (grazing), * indicates *p* < 0.05.

**Figure 2 biology-14-01143-f002:**
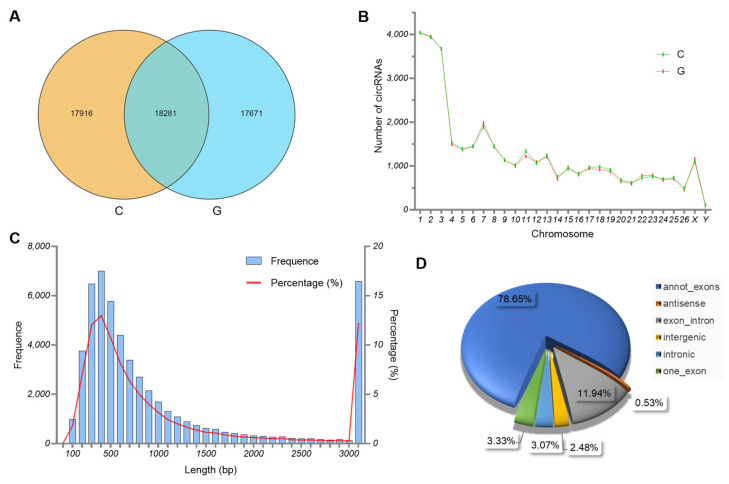
Characteristics of circRNAs in the biceps femoris tissues of Tibetan sheep from the C and G groups. (**A**) The number of expressed circRNAs. (**B**) The chromosomal distribution of circRNAs. (**C**) The length distribution of circRNAs. (**D**) The type of circRNAs.

**Figure 3 biology-14-01143-f003:**
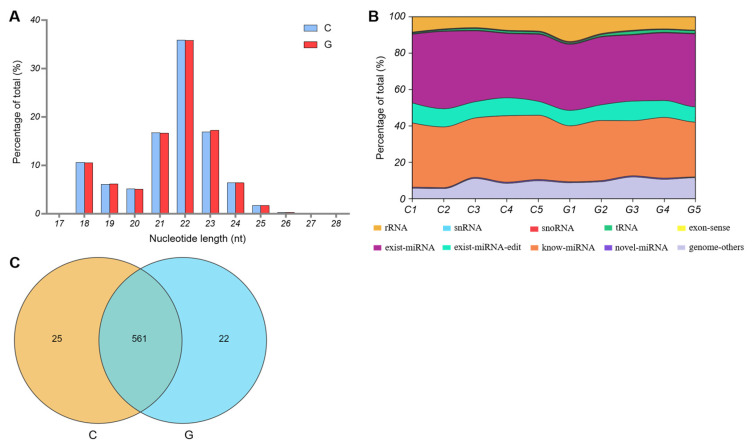
Characteristics of miRNAs in biceps femoris tissues of Tibetan sheep from C and G groups. (**A**) The small RNA length. (**B**) The small RNA types. (**C**) The number of expressed miRNAs.

**Figure 4 biology-14-01143-f004:**
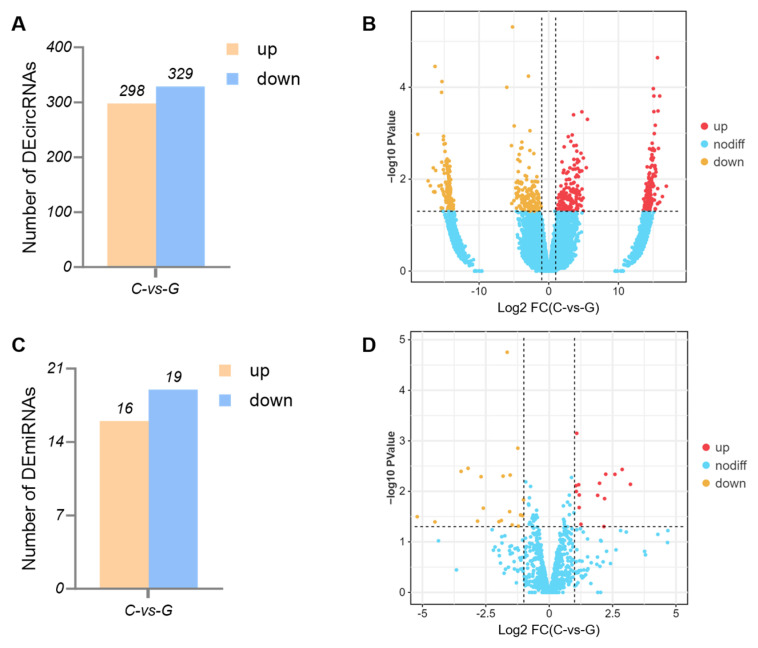
The differential expression of circRNAs (**A**,**B**) and miRNAs (**C**,**D**) in biceps femoris tissues of Tibetan sheep from C and G groups.

**Figure 5 biology-14-01143-f005:**
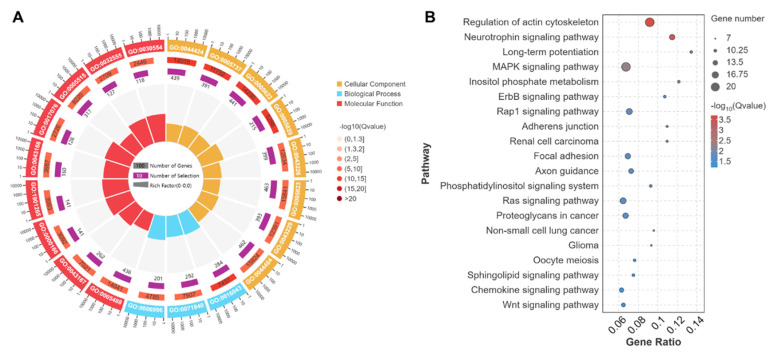
GO and KEGG analyses of source genes of DEcircRNAs. (**A**) Top 20 of annotated GO terms, (**B**) Top 20 of enriched KEGG pathways.

**Figure 6 biology-14-01143-f006:**
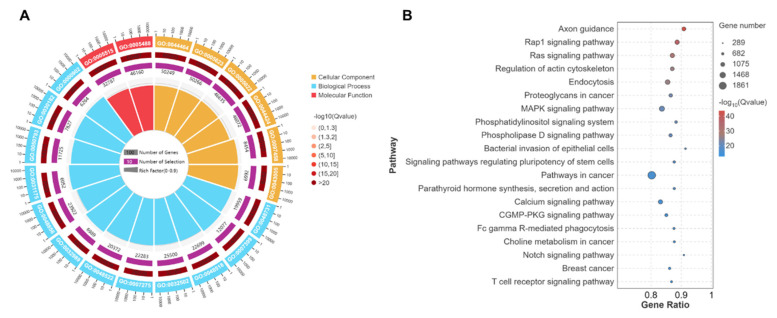
GO and KEGG analyses of target genes of DEmiRNAs. (**A**) Top 20 of annotated GO terms, (**B**) Top 20 of enriched KEGG pathways.

**Figure 7 biology-14-01143-f007:**
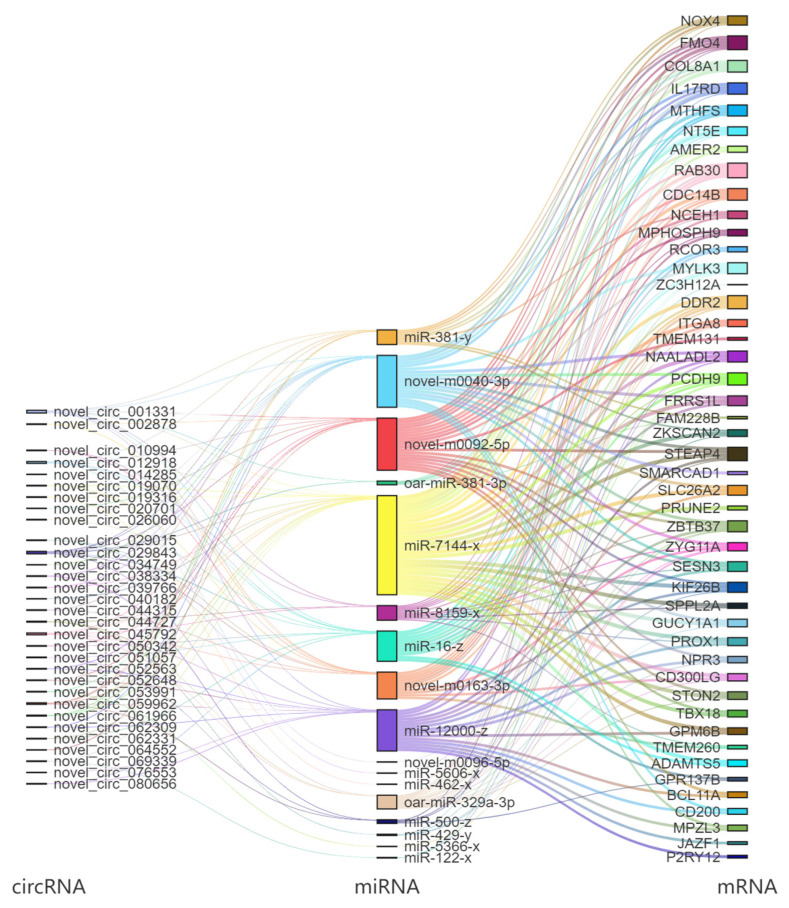
ceRNA Sankey diagram.

## Data Availability

The raw sequencing data used in this study were deposited in the NCBI Sequence Read Archive under BioProject accession number PRJNA1289012. All data generated or analyzed during this study are included in this published article.

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
