# Peer review of "Grazing-Induced Changes in circRNAs, miRNAs and mRNAs Expression in Tibetan Sheep Biceps Femoris"

_biology, 2025, doi:10.3390/biology14091143_

Round 1
Reviewer 1 Report
Comments and Suggestions for Authors
Dear Authors,
Congratulations on studying the molecular effects of exercise on meat quality in Tibetan sheep, focusing on the differences between natural grazing and stall-fed diets. Some of the corrections I've outlined below would be beneficial.
The historical and economic importance of the Tibetan sheep is well summarised. The literature support is adequate. However, similar expression studies in the literature can be compared and evaluated.
The exercise method is unclear in the material method section. The frequency, intensity and duration of the exercise can be briefly explained.
Line 94: The first two headings of the "Material and methods" section state that six animals from each group were studied. However, the third heading states that RNA was isolated from five animals from each group. Unless misspelled, the reason for this should be stated
Reviewer 2 Report
Comments and Suggestions for Authors
Comments about the abstract:
- The abstract lacks data, especially on how many animals were used.
- The term “exercise” is used ambiguously. In this context, “grazing” implies movement but is not a controlled exercise intervention.
-
The abstract suggests that exercise-mediated improvements in the meat quality of Tibetan sheep are "mediated through the circRNA–miRNA–mRNA regulatory network," specifically influencing muscle fiber type transformation and energy metabolism. Could the authors please clarify:
-
Which specific circRNAs, miRNAs, and mRNAs were identified as key regulators in this network? Additionally, please provide the fold changes in the expression levels of these RNAs compared with the control (pen-fed) sheep.
-
Which specific muscle fiber type transformation and energy metabolism pathways were found to be influenced by the circRNA–miRNA–mRNA regulatory network?
-
-
This study suggests that circRNA_39 and specific miRNAs may contribute to the improvement of meat quality in Tibetan sheep by regulating gene expression associated with muscle fiber type transformation and energy metabolism pathways. Could the authors please provide scientific evidence supporting this claim? Specifically:
-
What are the known or predicted functions of circRNA_39 and the miRNAs involved in this regulatory network?
-
Which downstream target genes or pathways (e.g., PGC-1α, MyHC isoforms, AMPK, or oxidative phosphorylation) were significantly upregulated in grazing sheep compared to pen-fed controls?
-
What were the fold changes in expression for these key genes, and how do these changes correlate with improved meat quality parameters (e.g., intramuscular fat content, fiber composition)?
-
What threshold (e.g., log2 fold change > 1 and adjusted p-value < 0.05) was used to define significant overexpression or differential expression?
-
- Consider replacing the exercise with grazing.
- Simple Summary: Please delete it, as it is not essential.
-
Could the authors please define what parameters were used to evaluate "meat quality" in this study? Specifically:
-
How was meat quality assessed and compared between grazing and pen-fed Tibetan sheep?
-
Which physical, chemical, or nutritional indicators (e.g., intramuscular fat content, protein concentration, fatty acid profile, tenderness, water-holding capacity, or color) were measured to reflect differences in meat quality?
-
What analytical or biochemical methods were used to determine these values (e.g., Soxhlet extraction for fat content, HPLC or GC-MS for fatty acids, or Warner-Bratzler shear force for tenderness)?
-
Based on these measurements, how was it concluded that meat from grazing sheep was of higher quality than that from pen-fed controls?
-
Comments about the introduction:
- Please add the reference to the following statement: ".... and is known as one of the three major primitive sheep breeds in China, together with the Kazakh sheep and Mongolian sheep (add reference).
- Lines 47-51: The sentence mixes biological/ecological information (population size, ecosystem stability) with sociocultural claims (livelihood, tradition) without separating them clearly. In scientific writing, it's essential to differentiate between ecological facts, socioeconomic relevance, and cultural importance—each requires separate justification and, ideally, separate citations.
- Lines 53-58: In meat science, muscle quality is evaluated by specific parameters: fibre composition, tenderness, water-holding capacity, pH, color, intramuscular fat, etc. Please specify what aspects of skeletal muscle quality are affected.
- Lines 58-66: Please revise the muscle fiber classification to reflect ruminant-specific types (Type I, IIa, IIx), as Type IIb fibers are generally absent in sheep. Clarify the metabolic roles of each fiber type and avoid ambiguous terms like "intermediate." Additionally, moderate the causal claims linking fiber types to meat quality and specify which traits (e.g., IMF, tenderness) were affected.
- The introduction outlines factors influencing meat quality but lacks a clear definition of what "meat quality" entails (e.g., IMF, tenderness, color). It would benefit from explicitly stating which specific meat quality traits are being evaluated and why they are important.
- Line 72-73: Please specify the intrinsic molecular mechanisms related to meat quality that were investigated in this study. How will changes in the transcriptome be used to evaluate meat quality traits? Additionally, please cite relevant studies, if available, where transcriptome analysis has been successfully used to assess meat or food quality in livestock or other food products.
Comments about the material and methods:
- Experimental Animals and Sample Collection: Please clarify how the exercise and control groups were defined, whether based on original feeding systems or assigned treatment. Ensure the euthanasia method meets ethical standards by specifying institutional approval. Also, provide more detail on the standardized diet used to control for nutritional differences.
- IMF Content Measurements: Specify the muscle sample preparation steps prior to Soxhlet extraction (e.g., drying or homogenization).
- RNA Extraction, Library Construction, and Sequencing: Please clarify why only five samples per group were sequenced, despite the six biological replicates mentioned earlier.
- Please provide the sequences of the 3′ and 5′ adaptors used for small RNA library preparation, or clearly state the specific library preparation kit and version employed.
- Read Mapping and Transcript Assembly: Authors should justify the use of default settings, or mention any specific parameter tuning if done.
- Identification of circRNAs and miRNAs: Authors should state whether RNA was treated with RNase R to remove linear RNAs, a standard step to enrich for circRNAs.
- Authors should describe the read alignment method used before circRNA detection, primarily whether the aligner supports circular RNA discovery.
- Authors should justify using miReap v0.20 or preferably use miRDeep2, the current standard.
- Quantification and Differential Expression Analysis: Please note that DESeq2 requires raw count data as input and performs its own normalization. Using RPM or TPM as input violates its statistical assumptions and may yield invalid results.
- Functional Annotation and Pathways Enrichment Analysis: Please clarify the thresholds and filtering criteria used for miRNA target prediction in miRanda and TargetScan.
- Construction of the circRNA-miRNA-mRNA Network: Line 158: Remove the following statement: "The ceRNA regulatory network was visualized by Sankey diagram."
- Please justify why different methods were used (e.g., "Pearson for linear expression trends, Spearman for monotonic but non-linear relationships.
Comments about the Results:
- Differences in Meat Quality: In this section, please specify the percentage increase in intramuscular fat (IMF) content in the exercise (E) group compared to the control (C) group. Additionally, briefly define how meat quality was evaluated—specifying which parameters were used (e.g., IMF content, tenderness, water-holding capacity, or others). Finally, explain how the differences in these parameters were quantified and, if applicable, how the overall improvement in meat quality was transformed into a percentage comparison between groups.
- Please add brief and informative descriptions (legends) to all figures in the manuscript to enhance the clarity and interpretation of the presented results. This will help readers understand the data more effectively and ensure that the figures are self-explanatory.
- Please check the Y-axis legend of figure 3A,B.
- Line 213: Please remove vague terms like "etc." and replace them with specific examples or a reference to supplementary data.
- Enhance the resolution of all figures of the manuscript and legends.
- Lines 264-265: Please correct this statement.
- 267-268: How does CircRNA significantly affect skeletal muscle quality and muscle fiber transformation? please explain
- In the discussion section, please elaborate on the specific changes in circRNA expression levels by quantifying how many fold the transcription of key circRNAs increased or decreased in relevant pathways compared with the control group. Additionally, discuss how these expression changes are linked to specific pathways (e.g., MAPK, Wnt, or muscle development-related processes) and how they contributed to the observed improvement in meat quality traits such as intramuscular fat (IMF) content, tenderness, or fiber composition. The current discussion is too generalized and would benefit from integrating specific transcriptomic data to support the proposed functional interpretations.
- The results and discussion sections should demonstrate which specific pathways and gene types are involved in the observed improvement in meat quality, and quantify how much their expression levels increased or decreased compared to the control group. This includes identifying key regulatory genes or pathways (e.g., MAPK, Wnt, metabolic, or muscle development-related genes) and linking their differential expression to meat quality traits such as intramuscular fat content, tenderness, or muscle fiber composition.
Comments about the Conclusion:
- The conclusion should highlight how the findings of this study can be applied to improve meat quality in Tibetan sheep through informed breeding, feeding, or management strategies.
There are several grammatical errors throughout the manuscript that need to be corrected through careful proofreading. The authors are encouraged to seek assistance from a language expert or professional editing service to improve the clarity and readability of their manuscript.
Round 2
Reviewer 2 Report
Comments and Suggestions for Authors
The authors have incorporated the suggestion recommended; therefore, the manuscript may be accepted